# SINE Insertion May Act as a Repressor to Affect the Expression of Pig *LEPROT* and Growth Traits

**DOI:** 10.3390/genes13081422

**Published:** 2022-08-10

**Authors:** Xiaoyan Wang, Chengling Chi, Jia He, Zhanyu Du, Yao Zheng, Enrico D’Alessandro, Cai Chen, Ali Shoaib Moawad, Emmanuel Asare, Chengyi Song

**Affiliations:** 1College of Animal Science & Technology, Yangzhou University, Yangzhou 225009, China; 2Department of Veterinary Science, Division of Animal Production, University of Messina, 98168 Messina, Italy; 3Department of Animal Production, Faculty of Agriculture, Kafrelsheikh University, Kafrelsheikh 33516, Egypt

**Keywords:** pig, RIPs, *LEPROT*, repressor, growth traits

## Abstract

Retrotransposon is an important component of the mammalian genome. Previous studies have shown that the expression of protein-coding genes was affected by the insertion of retrotransposon into the proximal genes, and the phenotype variations would be related to the retrotransposon insertion polymorphisms (RIPs). In this study, leptin (*LEP*), leptin receptor (*LEPR*), and leptin receptor overlapping transcript (*LEPROT*), which play important roles in the regulation of fat synthesis and body weight, were screened to search for the RIPs and their effect on phenotype and gene expression, as well as to further study the function of the insertion. The results showed that three RIPs located in intron 1 of *LEPROT* and intron 2 and 21 of *LEPR* were identified, and they were all SINEA1, which was one type of retrotransposon. The SINE insertion at the *LEPROT* was the dominant allele in native pig breeds. The age of 100 kg body weight of SINE+/+ Large White individuals was significantly higher than those of SINE+/− and SINE−/− individuals (*p* < 0.05). The *LEPROT* gene expression in the liver and suet of 30-day-old SINE−/− Sujiang piglets were significantly higher than those of SINE+/+ and SINE+/− piglets (*p* < 0.01). The dual-luciferase reporter gene assay showed that SINE insertion in PK15 and 3T3-L1 cells significantly reduced the promoter activity of the *LEPROT* gene (*p* < 0.01). Therefore, SINE insertion can be a repressor to reduce the expression of LEPROT and could be a useful molecular marker for assisted selection of growth traits in pig breeding.

## 1. Introduction

LEPTIN acts as a major regulator for food intake, body mass, energy homeostasis, and reproductive functioning [1,2,3]. It is secreted by fat cells and was first identified as the product of the obese (*OB*) gene and its effect on appetite and metabolic rate [4]. In vivo, LEPTIN must bind to its receptor (LEPTIN receptor, LEPR) to activate the downstream signaling pathway [5]. LEPR is a member of the class 1 cytokine family of receptors and is transcribed in localized areas in the brain and pituitary of the pig [6]. Leptin receptor overlapping transcript (LEPROT), which is also located upstream of *LEPR* in the pig genome, was first described as the product of an alternate protein using the same promoter of LEPR [7] and renamed Endospanin on account of its late endosomal/Golgi localization and membrane-spanning topology [8]. In mice, *LEPROT* knockdown showed high leptin sensitivity and was resistant to diet-induced obesity [9], and it was thought to be a candidate for regulating leptin sensitivity [10].

Numerous studies have been conducted on the association between polymorphism of *LEPTIN*, *LEPR*, *LEPROT*, and human obesity [2,11,12,13]. Consequently, the variation of these three genes related to domestic animals’ growth and carcass performance has been a major area of concern. The coding sequence of the *LEPROT* gene is affected by polymorphic simple sequence repeats (SSRs) with CA units ranging from 18 to 23 in miniature pig breeds compared with Duroc [14]. *LEPR* polymorphisms (1987C > T) were related to the fatness of Duroc finishing pigs [15]. Furthermore, using genome-wide association studies (GWAS), *LEPROT* and *LEPR* were strongly associated with bodyweight measured at 35-day-old broiler [16]. UASMS2 polymorphism in the *LEPTIN* gene was associated with average daily weight gain, body weight, and backfat thickness in 1378 cattle [17]. *LEPROT* also served as the candidate genes with biological support for ADG in pigs [18]. However, few studies have been reported about RIPs in these three genes.

Retrotransposons have been proven to be a component of the mammalian genome and have profoundly impacted the structure and function of the genome [19,20]. The insertion polymorphisms were reported to be related to the occurrence of diseases, such as cancer, autoimmunity, and neurodegeneration [21,22,23]. Retrotransposons include two major types: long terminal repeat (LTR) and non-LTR [24].

*Endogenous retroviruses* (*ERV*) are one of the superfamilies of mouse and human *LTR*s. *Long interspersed nuclear elements* (*LINE*s) and *short interspersed nuclear elements*
*(**SINE*s) are the main order of non-LTR retrotransposons [24]. Many researchers identified retrotransposons as cis-regulatory elements to modulate the expression of proximal or distal target genes [25,26]. Moreover, this may be the primary cause of phenotype variation by RIPs in domestic animals.

Recently, research works from our laboratory have indicated RIPs distribution in the pig genome [27,28]; their role in evaluating population structure [29,30]; and their insertion function on protein-coding genes and pig phenotypic variations such as *GH*, *TLR1*, and *TLR6* [31,32,33]. Therefore, considering the importance of *LEPTIN*, *LEPR*, and *LEPROT* on mammalian metabolism and growth, the RIPs in these three genes were identified, and the correlation between the RIPs and some pig economic performance was investigated in this study.

## 2. Materials and Methods

### 2.1. RIPs Screen and Identified

*LEPTIN* (ENSSSCG00000040464.2), *LEPR* (ENSSSCG00000025188.3), and *LEPROT* (ENSSSCG00000003806.4) and their flanking sequences (5 kb 5′ upstream and 3 kb 3′ downstream) were obtained from Ensembl (version Ensembl Release 101, http://asia.ensembl.org/index.html, accessed on 30 June 2020). As the bait, 3 genes in 15 assembled non-reference genomes (Landrace, Yorkshire, Pietrain, Berkshire, Hampshire, cross-breed of Yorkshire/Landrace/Duroc, Wuzhishan, Tibetan, Rongchang, Meishan, Bamei, Bama, Jinhua, Goettingen, and Ellegaard Gottingen minipigs) were downloaded from the NCBI database (version Annotation release 106, https://ncbi.nlm.nih.gov/, accessed on 1 July 2020) to screen for large structural variations (more than 50 bp) by sequence alignment using the ClustalX program. Retrotransposon insertion including SINE, LINE, and ERV among these predicted structural variations were annotated by RepeatMasker (version RepeatMasker 4.1.0 Released, http://www.repeatmasker.org/, accessed on 3 July 2020) with a customer-constructed library [28,32] and they were designated as RIPs. These RIPs were further validated by PCR amplification (Vazyme, Nanjing, China) in seven Chinese native pig breeds (Erhualian, Fengjing, Diannan small-ear, Wuzhishan, Bama, Tibetan, Meishan), three commercial pig breeds (Duroc, Landrace, Large White), one cross-breed (Sujiang), and wild boars. Two pooled DNA samples including six pigs for each breed were used. Total DNA was extracted from ear or blood tissues of each individual using the TIANamp Genomic DNA Kit (Tiangen, Beijing, China). Each DNA sample, after concentration detection by a spectrophotometer (NanoPhotometer N60 Touch, Implen Gmbh, Germany), was diluted to 50 ng/uL in concentration for pool mixture. Details of RIP evaluation for samples and primers are listed in Appendix A. All verified RIPs were sequenced at TsingKe Biological Technology Co. Ltd. (Nanjing, China).

### 2.2. RIP Genotyping

Nine breeds, namely, Duroc, Landrace, Large White, Erhualian, Bama, Jiangquhai, Jinghua, Wuzhishan, and Sujiang, which represented commercial pigs, Chinese native pigs, and cross pigs, were used to examine the RIP distribution. The individuals were selected according to their pedigrees, and all boars and sows without common ancestors were selected preferentially. The number of pigs used for each breed and the breed origins are listed in Appendix A. The genotype, the allele frequencies, and Hardy–Weinberg equilibrium were tested using Popgene [34]. Polymorphic information content (PIC) was calculated by the following formula:$$PIC=1-\overset{m}{\underset{i=1}{\sum }}P_{i}^{2}-\overset{m-1}{\underset{i=1}{\sum }}\overset{m}{\underset{j=i+1}{\sum }}2P_{i}^{2}P_{j}^{2}$$

### 2.3. Dual-Luciferase Reporter Assay

The promoter of *LEPROT* was predicted (version Promoter 2.0, https://services.healthtech.dtu.dk/service.php?Promoter-2.0, accessed on 25 September 2020), and one predicted region of *Leprot* with (852 bp) and without the SINE insertion allele (566 bp) was cloned from the Sujiang genomic DNA and verified by sequencing. The primers are listed in Appendix A. Then, the 852 bp and 566 bp sequences were inserted into the pGL3-basic vectors (Promega, Madison, WI, USA) to construct *LEPROT*SINE+-Luc+(En) vector and *LEPROT*SINE—Luc+ (En) vector, which contain β-globin and Oct4 mini promoters, respectively, for enhancer/repressor activity evaluation [32,35]. In 24-well plates, PK15 and 3T3-L1 cells were cultured to approximately 70% confluence and transfected with the luciferase reporter vector using the Lipofectamine 3000 reagent (Invitrogen, Carlsbad, CA, USA). After 48 h, luciferase activity was evaluated using the dual-luciferase reporter system (Promega, Madison, WI, USA) according to the manufacturer’s protocol.

### 2.4. Expression Analysis

Tissue samples including liver, suet, and backfat from four 30-day-old piglets for each genotype (SINE+/+, SINE+/−, and SINE−/−) were prepared. The mRNA was extracted using Trizol (Invitrogen, Shanghai, China), and 1000 ng RNA was reverse-transcribed into cDNAs using a TAKARA Kit (Takara, Tokyo, Japan). Then, quantitative real-time PCR (qPCR) was performed using the 7900HT Fast Real-Time PCR System (Applied Biosystems, New York, NY, USA) in a total volume of 20 μL containing SYBR mix (10 μL) (Takara, Tokyo, Japan), primers (4 ng), and cDNA sample (50 ng). Gene expression was normalized to glyceraldehyde-3-phosphate dehydrogenase (*GAPDH*) and measured using the 2− ΔΔCt method. The primer sequences also are shown in Appendix A.

### 2.5. Statistical Analysis

Statistical analyses were performed using the SPSS17.0 software package (SPSS, Chicago, IL, USA) using a one-way analysis of variance with Tukey’s post hoc test, and the data were expressed as mean ± SD. The records of body weight before slaughter, longissimus, and back fat thickness were collected in 450 Large White individuals from the Anhui Antai pig company, and the age at 100 kg body weight was adjusted using the formula recommended by the National Swine Genetic Assessment Scheme. ANOVA was used to analyze the associations between genotype and phenotype of body weight before slaughter, longissimus, and back fat thickness and the age at 100 kg body weight.

## 3. Results

### 3.1. Three RIPs Generated by Retrotransposon Insertions in the Pig LEPR and LEPROT Gene

After comparing *LEPTIN*, *LEPR,* and *LEPROT* genes in 1 reference genomes and 15 nonreference genomes using ClustalX, the polymorphisms of 54 structural variations that existed in more than one breed and over 50 bp were predicted and further annotated using RepeatMasker (Version 4.0.7) to find 32 predicted RIPs (Appendix A). Three polymorphic RIPs were identified in the DNA pool (Figure 1A) using PCR and gel electrophoresis. The diagrams of SINE insertion in these two genes and comparison among different species are shown in Figure 1B. They were all SINEA1, located in the first intron of *LEPROT*, and the second and 22 introns of *LEPR* gene that were 286 bp, 301 bp, and 321 bp, respectively (Table 1). They were assigned as *LEPROT*-SINE-RIP, *LEPR*-SINE-RIP1, and *LEPR*-SINE-RIP2.

### 3.2. Three RIP Distributions in Different Pig Breeds

Five Chinese fat-type pig breeds (Jiangquhai, Jinhua, Rongchang, Erhualian, and Bama), three commercial lean-type breeds (Large White, Duroc, and Landrace), and one dual-type breed (Sujiang pigs originated from hybrids of Duroc, Jiangquhai, and Fengjing) were selected to investigate the insertion allele frequency. The Hardy–Weinberg test and PIC value were also performed. According to Table 2, there was a difference between fat-type pig breeds and lean-type breeds for *LEPROT*-SINE-RIP. Insertion frequency was clearly low in Duroc and Landrace compared with in Jiangquhai, Jinghua, and Rongchang breeds. Sujiang pigs showed a similar tendency of allele frequency to commercial breeds. Three Chinese fat-type breeds were all in disequilibrium. For *LEPR*-SINE-RIP1, insertion frequency and the individual homologous insertion in all breeds were relatively low. SINE insertion was only found in Large White, Duroc, Sujiang, Jiangquhai, and Jinghua, and homologous insertion individuals only were found in Large White and Jinhua pigs. The insertion frequency of *LEPR*-SINE-RIP1 was low in Chinese native pigs compared with commercial pigs and cross pigs. There was no insertion homologous except Wuzhishan, and no obvious difference among Chinese native pigs and commercial pigs for *LEPR*-SINE-RIP2. PIC value indicated that all populations were low to moderately polymorphic.

### 3.3. Correlation of LEPROT-SINE-RIP with Growth Performance of Large White

Because of the distribution difference between Chinese native breeds and commercial breeds, the correlation of *LEPROT*-SINE-RIP and *LEPR*-SINE-RIP1 with the growth performance of Large White pigs was investigated (Table 3). The ages at 100 kg body weight of SINE insertion homologous were higher (*p* < 0.05) than heterozygote and homologous without SINE insertion, which indicated that SINE insertion homologous may take more 4 days to reach 100 kg bodyweight. There was no difference (*p* > 0.05) among the three genotypes of *LEPR*-SINE-RIP1 in the Large White population for other economic traits.

### 3.4. Expression Pattern of LEPROT in Tissues of 30-Day-Old Piglets

The *LEPROT* expression pattern was further investigated in the liver, suet, and back fat of 30-day-old piglets. As shown in Figure 2, the expression of SINE−/−genotype individuals was higher (*p* < 0.01) than those of SINE+/+ genotype individuals in the liver and suet, which indicated that SINE insertion may decrease the expression of *LEPROT* in these tissues.

### 3.5. SINE Insertion May Serve as a Repressor in Regulating the Expression of LEPROT

There was one predicted promoter that was 289 bp away from SINE insertion. One Duroc heterozygote was taken as a DNA template. The 878 bp sequences that contained predictor promoter and SINE insertion, and the 592 bp sequences that contained only the predicted promoter were cloned by PCR. These two fragments were inserted into the pGL3 vector. The schematic diagram of vectors is shown in Figure 3A. After sequencing and dual restriction enzyme digesting, the right vectors were named *LEPROT*SINE+-Luc+(EN) and *LEPROT*SINE--Luc+(EN). PK15 and 3T3 cells were transfected with these two vectors and the pGL3 control vector. The luciferase activity assay revealed that the SINE insertion significantly suppressed *LEPROT* gene promoter activities in PK15 and 3T3-L1 (*p* < 0.05) (Figure 3B). These data supported the SINE insertion that it may act as a repressor and could be involved in regulating *LEPROT*.

## 4. Discussion

In this study, two SINE insertion polymorphisms in *LEPR* and one in *LEPROT* were identified by PCR, since 32 RIPs were predicted by comparing these three genes (*LEPTIN*, *LEPR,* and *LEPROT*). According to the classification by Chen et al. (2019), these three SINE insertions belonged to the young retrotransposons subfamily. There was no identified RIP by PCR in the *LEPTIN* gene. LEPTIN acts as a multifunctional hormone cytokine and is involved in several cellular functions throughout the body. Through binding to LEPR, LEPTIN interacts with various signal transduction pathways such as JAK-STAT3 [36], PI3K/Akt [37], and MAPK [38], etc. Perhaps, because of Leptin’s significant role in metabolism and other biological processes, retrotransposon is limited not to propagating in this gene according to natural selection. Similarly, our previous study by Chen et al. (2021) on the growth hormone gene (*GH* gene), which encodes a pleiotropic hormone with important roles in promoting growth, anabolic actions, and body maintenance, had no RIPs identified.

Three RIPs’ distributions in different pig breeds, including Chinese native breeds and commercial pigs, were investigated. Insertion frequency was different between two types of pig and dominant for *LEPROT*-SINE-RIP in Chinese native breeds and *LEPR*-SINE-RIP in commercial pigs. Several studies have indicated that compared to commercial pigs, native pigs showed a slower growth rate, thicker backfat, and low lean percentage [39,40,41,42,43]. Natural selection and genetic drift are also powerful forces shaping the distribution and accumulation of retrotransposons [44]. Moreover, retrotransposon insertion has contributed to a selective advantage [45]. During the breeding of commercial pigs such as Large White and Duroc, intense selective pressure on lean percentage may change the insertion frequencies of RIPs in important protein coding genes related to myogenesis and adipogenesis, which may contribute to difference in RIP distribution among native pigs and commercial pigs. In a previous study, the difference in RIP distribution among Chinese native pigs and commercial pigs were identified [30]. Considering that the variation in *LEPROT* and *LEPR* was related to growth and fat deposition, the correlation between these two RIPs and some growth and carcass traits of Large White were further identified. The age of 100 kg body weight of SINE+/+ individuals was significantly higher than that of SINE+/− and SINE−/− individuals (*p* < 0.05), which indicated the SINE insertion homologous grew slower than the other genotypic individuals for *LEPROT*-SINE-RIP. There was no significant difference in backfat thickness among the three genotype individuals. Therefore, *LEPROT*-SINE-RIP could serve as a useful marker in selecting faster-growing pigs.

LEPROT, also named Endospanin, is a candidate for regulating LEPTIN sensitivity [10]. The expression of *LEPROT* was negatively correlated with LEPTIN receptors and LEPTIN sensitivity [46]. Schwartz et al. (1996) indicated that higher serum and plasma *LEPTIN* levels contributed to a higher total body fat percentage. Conversely, fat type pigs express higher levels of LEP*IN* mRNA and protein than lean-type pigs at similar body weight [47]. The amounts of leptin at plasma were higher in Meishan than Large White sows [48]. Diet-induced adiposity in genetically lean pigs also increases the leptin concentration in plasma [49]. However, the expression of *LEPROT* in the liver and suet of SINE−/− individuals in the present study were higher in subjects with relatively lower *LEPTIN* levels. This could have been a result of the deletion frequency being dominant for *LEPROT*-SINE-RIP in commercial breeds, which had a lower percentage of total body fat.

At last, the SINE insertion activity on the promoter of *LEPROT* was analyzed. Whether in PK15 or 3T3-L1, SINE insertion repressed the activity of *LEPROT* promoter. As a cis-regulatory element, SINE may regulate gene expression, acting as an enhancer or promoter [50,51]. The sequences provide abundant TF binding sites such as CTCF and ADNP [51]. When they escape silencing, they could be bound by multiple transcription factors (TFs) to acquire features of the bona fide enhancer or promoter elements and in turn exert a functional impact on gene regulation [52]. However, retrotransposon could be silenced by epigenetic regulation to ensure genomic integrity, leading to the repression of nearby genes. This might be due to the spreading of epigenetic marks and certain genes maintained in a lowly expressed state in specific development [53]. It is contradictory to understand the function of enhancers or repressors acted upon by retrotransposon. Moreover, their function may relate to specific cell fate, which could be shown in that different luciferase activities in different cell lines used the pGL3 report vector with the same SINE fragment in this study and our previous study. Considering that the expression of *LEPROT* in liver and suet of SINE−/− individuals were higher than that of the other two genotypes, SINE insertion in the first intron of *LEPROT* could have acted as a repressor to decrease the expression of *LERROT* and affect the growth traits of Large White pigs.

## 5. Conclusions

In summary, three RIPs were identified in intron 1 of *LEPROT* and intron 2 and 21 of *LEPR* by gene sequence alignment, retrotransposon annotation, and PCR verification. The distribution of *LEPROT*-SINE-RIP was relatively high in Chinese native pigs compared with commercial pigs. SINE insertion homologous may take more 4 days to reach 100 kg bodyweight than other two genotype individuals. Additionally, the *LEPROT* gene expression in the liver and suet of 30-day SINE−/− Sujiang piglets were significantly higher than those of SINE+/+ and SINE+/− piglets (*p* < 0.01). SINE insertion in PK15 and 3T3-L1 cells significantly reduced the promoter activity of the *LEPROT* gene (*p* < 0.01) using dual-luciferase reporter gene assay. Therefore, SINE insertion can be a repressor to reduce the expression of *LEPROT* and could be a useful molecular marker for assisted selection of growth traits in pig breeding.

## Figures and Tables

**Figure 1 genes-13-01422-f001:**
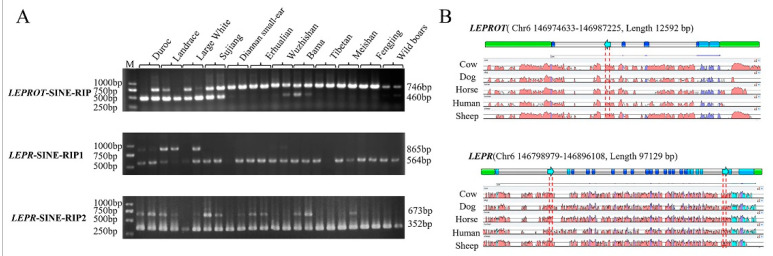
RIP identification and insertion loci diagram in *LEPROT* and *LEPR*. (**A**) The electrophoretogram of PCR production of three loci in *LEPROT* and *LEPR* genes in 12 breeds or population DNA pool. (**B**) Insertion loci diagram and sequence alignment for different species in *LEPROT* and *LEPR*.

**Figure 2 genes-13-01422-f002:**
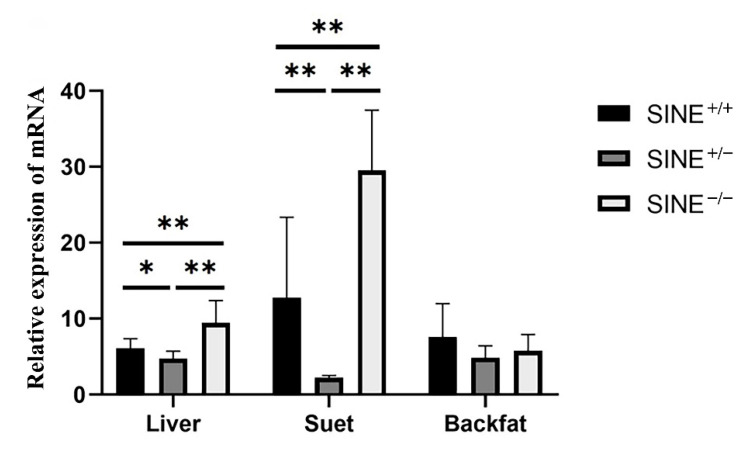
Relative expression of *LEPROT* gene in tissues of 30-day-old Sujiang piglets. * showed *p* < 0.05; ** showed *p* < 0.01.

**Figure 3 genes-13-01422-f003:**
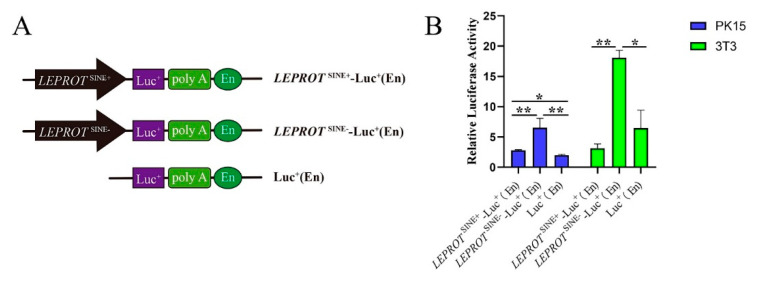
SINE insertion served as a repressor to affect the promoter of *LEPROT*. (**A**) Schematic diagram of the recombinant vector structure. (**B**) The results of promoter activity verification. * showed *p* < 0.05; ** showed *p* < 0.01.

**Table 1 genes-13-01422-t001:** The details of three RIPs in *LEPROT* and *LEPR* genes.

Loci	Insertion in Exon/Intron/UTR	Length (bp)	Direction	Type	Chromosome (Sus Scrofa11.1)
*LEPROT*-SINE-RIP	Intron1	286	-	SINEA1	6:146978072-146978073
*LEPR*-SINE-RIP1	Intron2	301	+	SINEA1	6:146873751-146874052
*LEPR*-SINE-RIP2	Intron21	321	+	SINEA1	6:146808226-146808547

**Table 2 genes-13-01422-t002:** Genotype and allele frequency of three RIPs in the RIP polymorphic breeds.

Polymorphic Site	Breeds	Number	Genotype Frequency			Allele Frequency		Hardy–Weinberg Equilibrium	Polymorphic Information Content
			+/+	+/−	−/−	+	−		
*LEPROT*-SINE-RIP	Large White	450	14.00	66.00	20.00	47.00	53.00	<0.01	0.374
	Duroc	24	12.50	66.67	20.83	45.83	54.17	0.09	0.373
	Landrace	24	0	37.50	62.50	18.75	81.25	0.26	0.258
	Sujiang	24	29.17	37.50	33.33	47.92	52.08	0.22	0.375
	Jiangquhai	24	62.50	20.83	16.67	72.92	27.08	0.02	0.317
	Jinhua	24	75.00	0	25.00	75.00	25.00	<0.01	0.305
	Rongchang	24	29.17	70.83	0	64.58	35.42	0.01	0.353
*LEPR*-SINE-RIP1	Large White	429	4.90	33.57	61.54	21.68	78.32	0.81	0.282
	Duroc	24	0	45.83	54.17	22.92	77.08	0.15	0.291
	Sujiang	24	0	62.50	37.50	31.25	58.75	0.03	0.490
	Jiangquhai	18	0	16.67	83.33	8.33	91.67	0.66	0.141
	Jinhua	24	4.17	29.17	66.67	18.75	81.25	0.83	0.258
*LEPR*-SINE-RIP2	Large White	429	0	62.70	37.30	31.35	68.65	<0.01	0.338
	Duroc	24	0	75.00	25.00	37.50	62.50	0.003	0.359
	Landrace	24	0	95.83	4.17	47.92	52.08	<0.01	0.375
	Sujiang	24	0	66.67	33.33	33.33	66.67	0.01	0.346
	Bama	24	0	87.50	12.50	43.75	56.25	0.0001	0.371
	Jiangquhai	24	0	33.33	66.67	16.67	83.33	0.33	0.239
	Erhualian	24	0	8.33	91.67	4.17	95.83	0.83	0.077
	Jinhua	24	0	37.50	62.50	18.75	81.25	0.26	0.258
	Wuzhishan	24	25.00	58.33	16.67	54.17	45.83	0.39	0.373

**Table 3 genes-13-01422-t003:** Correlation between *LEPROT*-SINE-RIP polymorphism and growth traits of Large White pigs.

Genotype	Number	Body Weight before Slaughter	Thickness of Backfat	Age at 100 kg Body Weight	100 kg Day-Old Backfat Thickness
SINE+/+	63	99.23 ± 1.26 ^A^	11.17 ± 0.37	164.81 ± 1.12 ^a^	11.20 ± 0.32
SINE+/−	297	105.10 ± 0.66 ^B^	11.53 ± 0.18	160.49 ± 0.59 ^B^	11.02 ± 0.15
SINE−/−	90	104.04 ± 1.20 ^B^	11.64 ± 0.34	160.59 ± 1.08 ^b^	11.22 ± 0.30

Note: Different superscript lowercase letters indicated difference between groups (*p* < 0.05). Different superscript capital letters indicated significant difference between groups (*p* < 0.01).

## Data Availability

All data needed to evaluate the conclusions in this paper are present either in the main text or the Appendix A.

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
