# Peer review of "SINE Insertion May Act as a Repressor to Affect the Expression of Pig LEPROT and Growth Traits"

_genes, 2022, doi:10.3390/genes13081422_

Round 1
Reviewer 1 Report
Dear Editor
I have reviewed the manuscript entitled ''SINE insertion may act as a repressor to affect the expression of pig LEPROT and growth traits''
The authors may give detailed information in material and methods about the reltionship of individuals they have choosen for each breeds since number of indivials are relatively low.
Similaryl they may discuss why some of the genotypes are low/high/absent in some breeds.
The english language may be checked.
best regards
Reviewer 2 Report
Additional reference sources primarily related to local pig breeds should be included.
Also, technical editing (during proof reading) is necessary for better presentation.
